# Balance-Associated Tests Contribute to Predicting the Need for Ambulatory Assistive Devices (AAD) among Community-Dwelling Older Adults

**DOI:** 10.3390/healthcare11172405

**Published:** 2023-08-28

**Authors:** Shiqi Xu, Lei Qian, Jianru Hao, Jun Wang, Yuyu Qiu

**Affiliations:** 1Wuxi School of Medicine, Jiangnan University, Wuxi 214126, China; 6212807045@stu.jiangnan.edu.cn (S.X.); 6222807021@stu.jiangnan.edu.cn (J.H.); 2Department of Rehabilitation, Wuxi 9th Affiliated Hospital of Soochow University, Wuxi 214023, China; qianleisure1985@gmail.com

**Keywords:** older adults, ambulatory assistive devices, balance associated tests, balance function

## Abstract

This study aims to analyze the use of ambulatory assistive devices (AAD) in relation to balance-associated tests and assist medical staff in providing professional objective reference values for older adults on whether to use AAD. Older adults (*n* = 228) were recruited from the local community to participate in this study. Participants were divided into the AAD-use group and the non-AAD-use group. Four balance-associated tests and scales were applied to predict the relationship between balance function and the use of AAD in older adults. They were used to assess the participant’s balance function and confidence in maintaining balance and were considered the most reliable measures of balance. There were significant differences in the Berg Balance Scale (BBS) score and Timed Up and Go Test (TUGT) among the subjects in the AAD-use group and non-AAD-use group (*p* < 0.001). The ROC curve analysis presented the following cut-off values for balance tests and scales: 23.62 s for the TUGT test and 41.5 points for the BBS score. For example, if the TUGT score is greater than 23.62 s and the BBS score is below 41.5 points, AAD is recommended for older adults to maintain balance and prevent falls. These objective reference standards may be useful in guiding medical personnel to determine whether older adults need to use AAD. In future studies, we hope to include more participants for subgroup analysis, investigating different types of AAD and their effects on older adults.

## 1. Introduction

Falls and injuries resulting from falls are a serious healthcare problem due to their association with subsequent morbidity, disability, hospitalization, institutionalization, and mortality [1,2]. They occur annually in 30% of adults over the age of 65 years [3]. By 2050, people older than 65 years are estimated to comprise 16% of the population [4]. With the increase in the population of older adults, more individuals will be at risk of falling [5]. Because of its serious consequences, the prevalence of falls in older adults has become a public health issue, which puts a great burden on society, families, and individuals [6]. The incidence of falls in older adults is highly related to the interaction of many factors, including decreased physical function, decreased balance function, disease, drug use, environmental factors, psychological status, and lack of care [7]. 

Ambulatory assistive devices (AAD), such as canes and walkers, are commonly used by those with walking limitations to sustain balance, increase mobility, and reduce fall risk [8]. A study reported that 24% of adults aged 65 and older used mobility devices in 2011 [9]. Although the use of such devices helps individuals to accomplish difficult activities and increases their participation in activities [10], it raises some safety concerns. Compared with non-AAD users, those who use AAD tend to have more falls [11,12,13,14]. In one study, approximately 22% of older adults using AAD reported falls [13]. Pellicer-García et al. indicated that using AAD was strongly associated with the risk of falling in older adults [15]. Roman et al. revealed that 68.2% of AAD users had experienced at least one fall [13]. One possible explanation for this is that the assistive device itself directly increases the risk of falls. When the device is lifted and advanced, biomechanical forces can be destabilized, and the balance may be disturbed by the need to allocate attention to the control of the device [16].

Balance and activity screening are the basic components of evidence-based clinical practice guidelines for fall prevention in older adults in the community [17]. Based on reference to existing studies, combined with the “World Guidelines for Falls Prevention and Management for Older Adults: A Global Initiative [18]”, the following indicators were determined as outcome indicators to judge the falling behavior of older adults. (1) The Modified Falls Efficacy Scale (MFES) can reflect the confidence of older adults in maintaining self-balance [19], and it has shown great convergent and structural validity [20]. (2) The Berg Balance Scale (BBS) that is applied broadly in clinics has generally been compared with other balance tests, and it is recognized to be the most reliable test for measuring balance [21,22]. (3) The Timed Up and Go Test (TUGT) has been suggested for evaluating gait and balance, and it has been verified as an appropriate measure for predicting the risk of falling in older adults [23]. (4) The One-legged Standing (OLS) test has been used specifically for static balance training and testing [24]. 

For the issue of whether older adults need to use AAD, at present, clinicians make subjective judgments based on their own clinical experience, and there is no objective index as a unified standard for reference. However, due to the different levels of clinical experience of medical personnel, the judgment on whether older adults need to use AAD is not uniform, which may lead to the misuse or absence of AAD, resulting in adverse consequences such as tendinitis or falls [16]. Therefore, our objective was to analyze differences in the balance function between the AAD-use group and non-AAD-use group, to help medical personnel provide professional objective reference values for older adults when determining whether or not to use AAD. Thus, this can save considerable time for medical staff and improve their work efficiency.

## 2. Materials and Methods

### 2.1. Study Design and Participants

This study was a descriptive cross-sectional survey to explore the differences between the AAD-use group and the non-AAD-use group. In total, 243 older adults were recruited from community healthcare centers in Wuxi between July 2021 and April 2022. 

The inclusion criteria were (1) age of ≥65 years, (2) ability to complete the tests included in this study, (3) ability to understand and follow instructions and (4) use of AAD under the professional evaluation of the medical personnel (based on their own clinical experience) if participants were using AAD. The following categories were excluded from our study: (1) individuals with a congenital physical disability, (2) individuals with recent surgical history (≤1 year) and (3) individuals with neurological, cardiopulmonary, or motor diseases that restrict walking.

Participants were divided into the AAD-use group and the non-AAD-use group. Apart from 12 individuals who were excluded because of various exclusion criteria, there were 126 participants eligible in the AAD-use group and another 105 in the non-AAD-use group.

The study protocol was approved by the Medical Ethics Committee of Jiangnan University (No. JNU20200717RB02). All participants signed a consent form. We encouraged nurses and physicians to attend meetings to explain the purpose of the study, as these are the reference professionals for community-dwelling older adults and could inform potential participants to participate in the study. Posters were also made and placed in waiting rooms at primary health care and community social centers, informing both the older adults attending the center and their relatives of the aims of the study.

### 2.2. Data Collection

General information about the subjects was collected with their consent, including gender, age, height, weight, disease history, etc. Demographics and clinical data (including balance tests) were collected by the same researcher. Interviews were also conducted with physiotherapists to verify whether or not AAD was appropriate for the participants’ individual needs.

The subjects were informed of the test contents and methods in detail, which aimed to make sure all subjects became familiar with the whole test process and implemented it as required. Before the test was performed, the subjects were required to complete appropriate warm-up exercises. Sufficient rest time was available for each subject in the interval between the two test items. There were two staff members providing safeguard procedures throughout the whole process in case of unforeseen circumstances such as falls.

### 2.3. Balance Tests

#### 2.3.1. Modified Falls Efficacy Scale (MFES)

The MFES contains 9 indoor activity tests and 5 outdoor activity tests [19]. The nine indoor activity tests are respectively dressing or undressing, preparing simple meals, taking a bath, sitting or rising in a chair, getting up or going to bed, opening the door or answering the phone, walking around the room, reaching into the cupboard or wardrobe, and light physical housework. The five outdoor activity tests include the use of public transport, road crossing, light gardening or hanging clothes, simple shopping, and using steps [25]. Each of these 14 tests is scored from 0 (no confidence at all) to 10 (sufficient confidence) with 11 grades [19,25]. 

#### 2.3.2. Berg Balance Scale (BBS)

The BBS is an instrument that assesses balance function in older adults using 14 tasks: sitting to standing, standing unaided, sitting unaided, standing to sitting, transferring, standing with eyes closed, standing with feet together, reaching forward with an outstretched arm, retrieving an entity from the floor, turning to look behind, turning 360 degrees, placing the foot alternately on a step, standing with one foot in front, and standing on one foot. The maximum score is 56 points, representing normal balance. Each item is scored from 0 (unable to perform) to 4 (general performance) [22,26]. The lower the score, the more likely the balance function is worse [22]. 

#### 2.3.3. Timed Up and Go Test (TUGT)

The TUGT is one of the fall screening tools identified in foreign clinical guidelines as an appropriate screening tool and is widely used to assess the risk of falls in older adults [27]. A staff member prepares a chair with a height of about 45 cm and armrests of 20 cm and makes a clear mark on the floor 3 m away from the seat. The researcher measures the time for a participant to get off the chair, walk 3 m at their ordinary speed through the marked line, then turn around, walk back, and sit on the chair. This test is repeated three times, with the average as the final result [28].

#### 2.3.4. The One-Legged Standing (OLS)

OLS is a tool for forecasting frailty in community-dwelling older adults. The subject’s dominant foot stands on the flat ground, and the non-dominant foot rises about 20~30 cm without leaning on the supporting leg, and then looks straight ahead [24]. The researchers record the time that the subjects maintain this position. Two staff members complete the test, respectively, implementing timing and standing beside the subjects to protect them from accidental falls. 

### 2.4. Statistical Analysis

All statistical analysis was conducted using IBM SPSS Statistics version 20.0 (IBM Corporation, Armonk, NY, USA). The general data and evaluation results of the two groups were inputted, and a series of statistical analyses were carried out. Differences between variables were evaluated using the independent sample T-test (continuous variables) or chi-squared test (categorical variables). Continuous variables were presented as mean ± SD with categorical variables as an absolute number and percentage (%) of the total. Pearson correlation analysis was performed to examine the relationship between indicators of balance function and the usage of AAD by older adults. We used logistic regression to build models with whether older adults need to use AAD as the dependent variable and four balance-associated tests and scales as the independent variables to screen out the influence variables of AAD use in the older adults. Taking the screened variables as the research object, ROC curves were established to analyze sensitivity and specificity. The sensitivity of a test showed how well it correctly identified subjects with a condition of interest, whereas specificity illustrated the frequency that the test was negative in the absence of a condition of interest [29]. We calculated the best critical value of this variable to predict if older adults need to use AAD. The level of significance was set at 0.05.

## 3. Results

### 3.1. Participant Characteristics

At the beginning of our study, 243 older adults in community healthcare centers were recruited (Figure 1). Twelve individuals were excluded because they met various exclusion criteria: three older adults presented with a congenital physical disability, three older adults recently had a surgical history (≤1 year), and six older adults suffered cardiopulmonary disease that restricted walking. During the study procedure, 3 older adults withdrew and could not complete the tests due to fragility. 

Of the 228 subjects in our sample, 102 subjects (45 females and 57 males) belonged to the non-AAD-use group and 126 subjects (66 females and 60 males) belonged to the AAD-use group. All 228 subjects completed the scales and tests. The demographic and clinical characteristics of the two groups are shown in Table 1. Gender, body height, body weight, body mass index (BMI), heart disease, diabetes, lower limb arthritis, and Parkinson’s disease showed no statistically significant differences between the two groups. However, the results of age, hypertension, MFES score, BBS score, TUGT, and OLS test showed significant differences between the AAD-use group and non-AAD-use group (*p* < 0.05). 

### 3.2. Correlation between Tests and the Use of AAD

Furthermore, Table 2 shows a significant correlation between the MFES score, BBS score, TUGT, OLS test, and the use of AAD based on Pearson correlation analysis. The results showed a strongly negative correlation between MFES, BBS, OLS and the use of AAD with the score of −0.393, −0.434 and −0.140, respectively, while a strongly positive correlation was shown between the TUGT and the use of AAD with a score of 0.779.

As presented in Table 3, whether to use AAD was set as a dependent variable (non-AAD-use defined as 0, AAD-use defined as 1). The results showed that the TUGT, BBS test, and OLS test, among the four tests, could be used as indexes to judge whether to use AAD or not. In the TUGT, the influence coefficient was 1.426, which meant that the longer time consumed the higher possibility of using AAD. In contrast, in the BBS test and OLS test, the influence coefficient was −0.336, −0.940, respectively, which meant that the lower score was achieved with the higher possibility of using AAD.

### 3.3. Ability to Predict the Application of AAD

The ROC curve was used to analyze the results of the TUGT, BBS test, and OLS test in order to determine whether older adults need to use AAD. No significant differences between the groups were found in the sensitivity and specificity of OLS test. Whether older adults need to use AAD is strongly affected by the TUGT and BBS test. As shown in Figure 2, the AUC of the TUGT was 0.974 (95% CI 0.955–0.993), which showed an excellent predictive ability. The formula for calculating the Youden index is as follows:Youden index = sensitivity + specificity − 1,(1)

Based on the maximum value of the Youden index, the corresponding critical value was 23.62 s, with sensitivity (97.62%) and specificity (91.18%). The AUC of the BBS test was 0.755 (95% CI 0.693–0.817), and this also meant a good predictive ability. The optimal cut-off score maximizing sensitivity (60.32%) and specificity (81.37%) on the curve was a BBS score of 41.5.

## 4. Discussion

Older adults with a high risk of falls have attracted particular attention in relation to frailty among the older adult population. A fall can cause various types of injuries such as bone fractures and craniocerebral injuries, resulting in limited movement, long-term bedridden ness, lower quality of life, and even disability or death [30]. In the past few decades, agreement has been reached that the use of AAD is one of the important measures to prevent and reduce falls in older adults [31]. Nevertheless, some studies have found that AAD can also lead to falls. Cruz AO reported that fall incidence was more frequent in frail older adults with AAD dependency than in controls in the same state of weakness [14]. This could be explained by the fact that the assistive device itself directly increases the risk of falls. As the device is raised and advanced, biomechanical forces may be destabilized, and balance may be disrupted by the need to allocate attention to controlling the device. Among older adults aged 65 and older, more than half of the AAD users were influenced by their family and friends to make decisions to use AAD without professional guidance [32]. The lack of professional guidance leads to a high incidence of falls among AAD users [33]. Although Joo et al. found that the 10-s tandem stance test may be useful in guiding mobility aid prescription, the sample size was too small and lacked comparison with criterion standards such as the BBS test [34].To date, there is no gold standard screening for addressing the need for AAD among older adults in the community. Thus, it is an urgent problem to help medical personnel provide professional objective reference values for older adults regarding whether or not to use AAD. 

The establishment of walking safety for older adults depends on their dynamic and static balance ability, psychology, lower limb muscle strength, movement coordination, comprehensive ability, and other factors [32]. The evaluation indexes used in this study cover the above aspects as much as possible. This study adopted a combination of subjective and objective methods to evaluate the balance ability of patients. The subjective evaluation tool was MFES, which was completed in the form of a questionnaire. From the current literature, the scale is mainly used for hospitalized older patients. With excellent AUC value, screening rate, predictive sensitivity, and specificity, MFES can be rapidly applied in clinics [35]. Many factors were taken into consideration when we selected the objective evaluation tools, such as ease of evaluation, time consumption, equipment use, funds, and so on. The BBS is a widely used balance function assessment scale at present, which can be used not only for various types of inpatients but also for older adults in the community with good reliability, validity, specificity, and predictability [36]. The TUGT includes walking and steering processes to assess the function of nerves and muscles, such as strength, balance, flexibility, and so on. It provides a more comprehensive and reliable assessment of fall risk in older adults, and some scholars believe that it is more effective in older adults with insufficient motor function [37]. The OLS test, which is widely applied in clinical settings, is usually used for evaluating the stability of standing posture in older adults. OLS is associated with age, self-assessment of health, body mass index, mortality, and fall risk [21]. It is also beneficial for identifying older adults with increased risk of functional dependence in the future [24]. 

In our study, there were significant differences in MFES, BBS score, TUGT evaluation, and the OLS test, which were correlated with the use of AAD. Our study indicated that there was a strongly positive correlation between the TUGT and the use of AAD, and a strongly negative correlation among the MFES score, BBS score, OLS score, and the use of AAD. Through regression analysis, it was found that the TUGT test, BBS scale, and OLS test played important roles in the judgment of whether older adults need an AAD. However, only the TUGT and BBS test showed significant differences in the sensitivity and specificity between the two groups. The study of Reuben et al. suggested that the completion time of the TUGT test in the Japanese older adult population was less than 20 s, indicating that they had independent activity ability. On the contrary, the cut-off time was more than 30 s with sensitivity (54%) and specificity (74%), suggesting that the subjects needed aids to complete most of the activities [38]. Jungui Zhou et al. pointed out that when the diagnostic cut-off value of BBS was 40 points, the sensitivity (68%) and specificity (57%) were the highest [39]. The results of the above studies are close to the optimal critical value determined by our study, and our results demonstrate higher sensitivity and specificity. 

Our finding that age was one of the influential factors in participants’ use of AAD is similar to the study conducted by Haibin Zhou et al. [40]. In this study, MFES scores, BBS scores, and OLS scores were found to have a strong negative correlation with the use of AAD. This means that the lower scores the subject achieved, the more he needs to use AAD. For the MFES, a lower score indicated more fear of falling. But based on the results of binary logistic regression analysis, MFES cannot be a significant index for determining whether older adults need an AAD or not. This could be due to its subjectivity. Meanwhile, older adults in particular have a limited understanding of option settings, which makes it difficult to accurately assess the need for AAD. Our study also proves the disadvantages of a subjective scale in practical application. The actual behavior of older adults with the use of AAD depends on a variety of factors, including their mental state, balance function, reduced muscle strength [41,42], response-ability, and so on. Relatively speaking, MFES focuses on the mental state of older adults, while the BBS and TUGT focus on the performance of balance in different movement states. Nevertheless, as shown in a previous study, the Timed Up and Go test had limited ability to predict falls in community-dwelling older adults, so we should not use the TUGT in isolation to identify individuals at a high risk of falls in this setting [43]. It was confirmed that the integration of several measures of postural stability can capture the multifactorial nature of fall risk better than a single test [44]. Therefore, in future research, we can take the TUGT test and BBS score together into consideration as the predicting indicators of whether older adults need to use AAD, and then verify the practical value. From a clinical point of view, medical personnel can combine their own clinical experience with objective indicators to identify the need for AAD for balance impairments and falls prevention. Such an approach has the potential to maximize efficiency, reduce assessment time, and increase the confidence of policymakers in recommending the use of AAD.

There are some limitations on this study that should be recognized. First, the difference in age is a major limitation in this study. There are two primary reasons for this. First one, the majority of people who use walkers are older adults. Another reason is that some patients were not included during the cross-sectional study due to personal reasons or inability to co-operate, leading to bias. We will concentrate on the relationship between age and the use of AAD in future studies. Second, the behavior of older adults using AAD is affected by many factors, which may include balance ability, physical fitness, mental state, weight, gender, cultural level, habits, and so on. All the factors associated cannot be absolutely included in this study. Finally, for each individual, the factors that play decisive roles are also inconsistent, and being limited by the research methods, this study is unable to study the weight of each influencing factor. In addition, compared with the influencing factors included in this study, the number of subjects recruited is moderately insufficient, and for this reason, the AAD used by the subjects could not be classified.

## 5. Conclusions

The MFES score and BBS score of older adults with AAD were lower, and the TUGT test was more time-consuming than that of older adults without AAD. Specifically, compared with those who did not use AAD, AAD-users had lower confidence in maintaining balance and a higher risk of falls. In addition, the TUGT test and BBS scores have better predictive abilities for the use of AAD in older adults, while the best critical value of the TUGT was 23.62 s, and of the BBS was 41.5 points. That is, when the TUGT test result was greater than 23.62 s and the BBS score was less than 41.5, older adults were advised to use AAD to maintain balance and prevent falls. These objective reference standards may be useful in guiding medical staff to determine whether older adults need to use AAD. However, this assumption still needs to be supported by further investigation. Given our results and those already available in the literature, we recommend that future studies in this area include a larger number of individuals for subgroup analysis, investigating different types of AAD and their effects on older adults.

## Figures and Tables

**Figure 1 healthcare-11-02405-f001:**
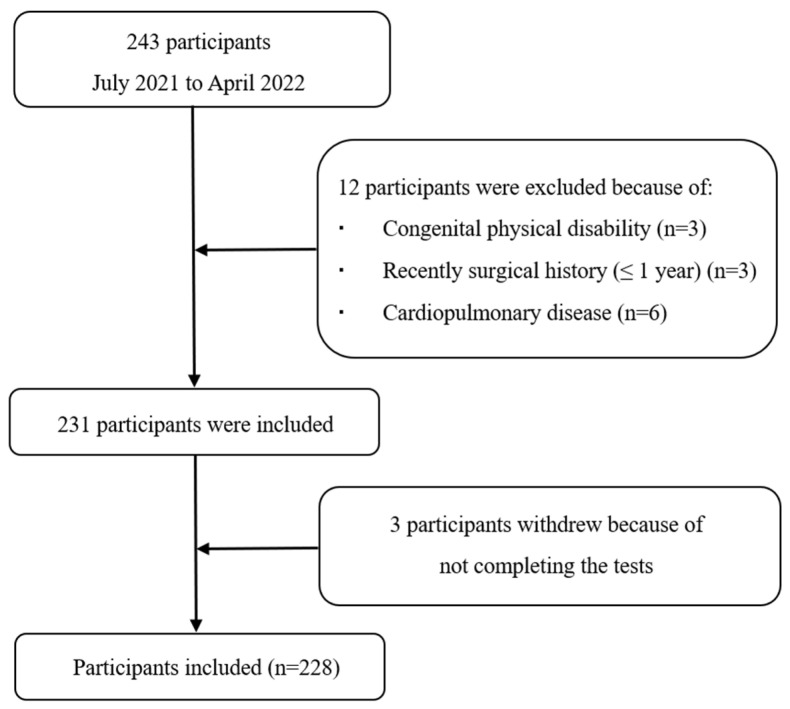
Flow chart of participants’ enrollment.

**Figure 2 healthcare-11-02405-f002:**
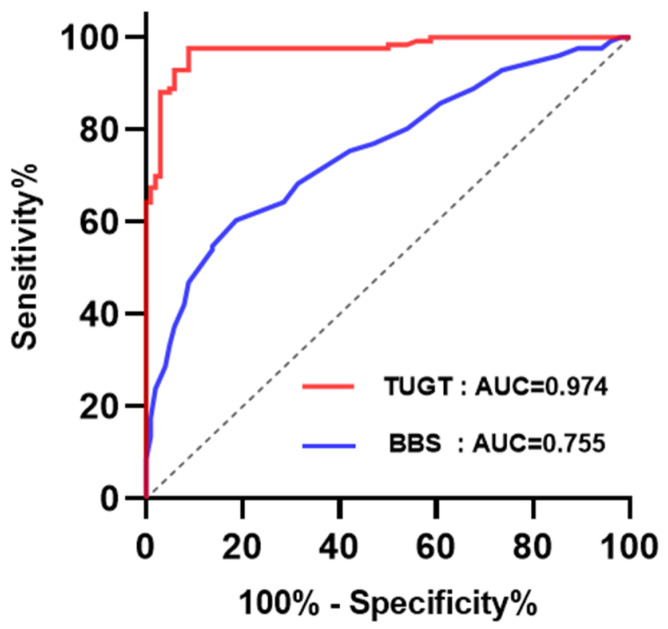
ROC curve of TUGT and BBS as predictors of the application of AAD for elders. Abbreviation: ROC, receiver operating characteristic; BBS, Berg Balance Scale; TUGT, Timed Up and Go Test.

**Table 1 healthcare-11-02405-t001:** Sample demographic characteristics (*n* = 228).

Variables	Non-AAD-Use Group (*n* = 102)	AAD-Use Group (*n* = 126)	t/χ^2^	*p*-Value
Gender (male, *n*, %)	45 (44.1)	66 (52.4)	1.541	0.215
Age (y, *n*, %)			22.818	<0.001
65–69	18 (17.6)	3 (2.4)		
70–79	15 (14.7)	24 (19.0)		
80–89	57 (55.9)	63 (50.0)		
≥90	12 (11.8)	36 (28.6)		
Height(m)	1.62 ± 0.10	1.64 ± 0.07	−1.613	0.108
Weight (kg)	63.40 ± 13.48	64.69 ± 10.05	−0.801	0.424
BMI (kg/m^2^)	23.96 ± 4.32	24.00 ± 3.60	−0.074	0.941
Hypertension (*n*, %)	48 (47.1)	81 (64.3)	6.809	0.009
Heart disease (*n*, %)	21 (20.6)	15 (11.9)	3.197	0.074
Diabetes (*n*, %)	15 (14.7)	30 (23.8)	2.949	0.086
Lower limb arthritis (*n*, %)	6 (5.9)	3 (2.4)	1.016	0.313
Parkinson’s Disease (*n*, %)	9 (8.8)	6 (4.8)	1.513	0.219
MFES (score)	101.11 ± 22.60	77.37 ± 31.26	6.643	<0.001
BBS (score)	45.76 ± 5.46	38.83 ± 8.33	7.547	<0.001
TUGT (s)	19.42 ± 3.25	32.82 ± 6.63	−19.921	<0.001
OLS (s)	6.39 ± 2.75	5.66 ± 2.41	2.131	0.034

Abbreviation: AAD, ambulatory assistive devices; BMI, body mass index; MFES, Modified Falls Efficacy Scale; BBS, Berg Balance Scale; TUGT, Timed Up and Go Test; OLS, The One-Legged Standing.

**Table 2 healthcare-11-02405-t002:** Correlation analysis between tests and the use of ambulatory assistive devices (AAD).

	Whether the AAD Was Used or Not	MFES	BBS	TUGT	OLS
Whether the AAD was used or not	1	-	-	-	-
MFES	−0.393 **	1	-	-	-
BBS	−0.434 **	0.350 **	1	-	-
TUGT	0.779 **	−0.472 **	−0.405 **	1	-
OLS	−0.140 *	0.017	0.143 *	0.078	1

Abbreviation: AAD, ambulatory assistive devices; MFES, Modified Falls Efficacy Scale; BBS, Berg Balance Scale; TUGT, Timed Up and Go Test; OLS, The One-Legged Standing. * indicates that the *p* value was less than 0.05, ** indicates that the *p* value was less than 0.01.

**Table 3 healthcare-11-02405-t003:** Multivariate logistic regression analysis of related indexes in subjects (*n* = 228).

Variables	B	*p*	OR	95% CI	
				Lower Limit	Upper Limit
Age	−1.789	0.159	0.167	0.043	0.644
Hypertension	−1.231	0.253	0.292	0.035	2.405
BBS	−0.336	0.010	0.714	0.553	0.922
TUGT	1.426	0.001	4.163	1.851	9.366
MFES	0.089	0.055	1.093	1.002	1.193
OLS	−0.940	0.003	0.391	0.212	0.720

Abbreviation: MFES, Modified Falls Efficacy Scale; BBS, Berg Balance Scale; TUGT, Timed Up and Go Test; OR: odds ratio; CI: confidence interval.

## Data Availability

Not applicable.

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
