# Peer review of "Balance-Associated Tests Contribute to Predicting the Need for Ambulatory Assistive Devices (AAD) among Community-Dwelling Older Adults"

_healthcare, 2023, doi:10.3390/healthcare11172405_

Round 1
Reviewer 1 Report
Overall, the scientific article is well-structured and presents valuable findings. Refining certain key aspects would help strengthen the quality and impact of the presented research.
Please avoid the use of elderly in the manuscript. Older adults is the correct form nowadays.
Tables and figures: please explain the acronyms.
Abstract:
The description of the balance-associated tests and scales used could be more detailed, including their specific relevance to balance assessment. Additionally, while the ROC curve analysis results are presented, the practical significance of the identified threshold values should be discussed further. Providing insights into the clinical implications of these findings and potential recommendations for healthcare providers would enhance the abstract's impact. Furthermore, a brief mention of the limitations or potential future directions of the study would be beneficial for a more comprehensive understanding
Introduction:
Organize the introduction with a logical flow. Start by outlining the growing concern of falls in the older population and the relevance of AAD in fall prevention. Then, transition into the specific challenges and gaps in AAD use, and finally, introduce the objective of the study as a response to these challenges.
"expert consensus on Fall Risk assessment for the Elderly in China (draft)": I think this is not the way to references this
The introduction should conclude by explicitly stating the study's objective and its significance. The current conclusion is somewhat abrupt and doesn't clearly tie back to the study's purpose
Material and Methods:
While the study design is described as a "descriptive cross-sectional survey," it would be beneficial to provide more details on the purpose and context of choosing this specific design. Clarifying how this design aligns with the study's objectives and why it's appropriate for investigating the relationship between AAD use and balance-associated tests would strengthen the rationale
Elaborate a bit more on the recruitment process
While the general data collected is briefly mentioned, it would be valuable to expand upon how the subjects' general information was collected and documented, ensuring that the data collection was standardized and consistent
“great potential safety hazards when walking” is a little bit open to be a exclusion criteria. Please explain this point.
Results:
it would be useful to include more context regarding their mean age, as age differences are mentioned as statistically significant. Providing this information would help readers understand the scale of the age difference between the AAD-use and non-AAD-use groups
The Multivariate logistic regression should include other statistical different variants: aged and hypertension.
Discussion:
Discuss the potential implications of the correlation between AAD use and falls, considering that AAD is used as both a preventive and potentially contributory measure.
When discussing the existing studies' results, provide a bit more context regarding their methodologies. Please comment this study in the discussion section: Joo B, Marquez JL, Osmotherly PG. Ten-Second Tandem Stance Test: A Potential Tool to Assist Walking Aid Prescription and Falls Risk in Balance Impaired Individuals. Arch Rehabil Res Clin Transl. 2021 Dec 6;4(1):100173. It will improve the findings.
The idea of combining TUGT and BBS as predictors of AAD need is promising. Expand upon the advantages of using a multifactorial approach and its potential benefits in clinical decision-making. Discuss how these findings could be practically applied by medical personnel
No limitations section: Focus more on the potential limitations of the study and how these limitations might affect the interpretation of the results and their clinical applicability. The different in aged is a mayor limitation in this study based on a miss the recruitment process. Please explain it
The conclusion could be enhanced by emphasizing the clinical significance of differences in MFES and BBS scores, as well as prolonged TUGT test duration among AAD users. Exploring practical implications and comparisons with existing fall risk standards would strengthen the findings' relevance. Additionally, discussing potential directions for future research, such as investigating specific AAD types and their impact on diverse elderly subgroups, could provide a more comprehensive perspective.
Author Response
Dear Reviewer:
Thanks a lot for your comments concerning our manuscript. Those comments are very valuable and helpful for revising and improving our paper. We have studied all comments carefully and have made conscientious correction. Revised portion are marked in the paper. The responds to comments are upload as a Word file. Please see the attachment.
Many thanks!
Best regards,
All authors

Reviewer 2 Report
HEALTHCARE-2550583 presents results for use of AAD in older adults. While some parts of the paper were interesting, other areas could be improved. I hope the authors consider my feedback.
MAJOR COMMENTS
· Avoid the use of “elderly” throughout. Instead “older adults”.
· Lines 31-40: The use of AAD and fall risk might be because AAD users have severe mobility limitations? This should be addressed somehow in the text.
· Section 2.1 needs far more detail.
· Section 2.2: Remove the bullet points and transform into paragraph form.
· Section 2.5: Can the authors defend the use of Pearson correlations instead of ICC or spearman in the text?
· Line 138: What is logical regression? Did you mean logistic?
MINOR COMMENTS
· Line 24: Insert “years” after “65”.
· Line 30: Delete, “and so on”.
· Table 1: Remove the X2 column and “*”.
· Table 2: Avoid abbreviations in the title and define all abbreviations in a table note.
· Lines 178-184: The <0 and >0 is not well understood. Just instead list the correlation coefficient.
· Line 246 and elsewhere: Remove all Results from the Discussion.
· Make any changes to the abstract that align with those from the text.
Minor grammatical errors need correction and the use of "elderly" need to change to "older adults" throughout.
Author Response

(The authors gave the same response as above.)

Round 2
Reviewer 1 Report
After the changes made, the article has significantly improved and is ready for publication. Congratulations to the authors. However, I believe a couple of minor changes can further enhance the article:
- Figure 1 still includes “elderly”
- While we cannot provide specific age values due to the design of your demographic questionnaire, it will important to discussing the age distribution and its potential impact on the study outcomes. Also expand this limitation in the discussion section, highlighting the importance of age as a factor of loss of balance and falls.
Author Response
Dear Reviewer,
Thanks again for your comments concerning our manuscript entitled “Balance-associated tests contribute to predicting the need for ambulatory assistive devices (AAD) among community-dwelling older adults” (healthcare-2550583). We have carefully studied all of the comments and made conscientious correction. The revised part is marked in the paper. Here are the responses to the comments.Please see the attachment.

Reviewer 2 Report
The authors did a nice job addressing my previous concerns, but I have a couple more points (related to the previous review):
***Line 82: "was" should be "were". Again, be sure to quality control minor English language edits.
***Figure 1 still states "elderly". Please review and replace "elderly" with "older adults" where appropriate.
Author Response

(The authors gave the same response as above.)
